# Mask-based Membership Inference Attacks for Retrieval-Augmented Generation

## ABSTRACT

Retrieval-Augmented Generation (RAG) has been an effective approach to mitigate hallucinations in large language models (LLMs) by incorporating up-to-date and domain-specific knowledge. Recently, there has been a trend of storing up-to-date or copyrighted data in RAG knowledge databases instead of using it for LLM training. This practice has raised concerns about Membership Inference Attacks (MIAs), which aim to detect if a specific target document is stored in the RAG system's knowledge database so as to protect the rights of data producers. While research has focused on enhancing the trustworthiness of RAG systems, existing MIAs for RAG systems remain largely insufficient. Previous work either relies solely on the RAG system's judgment or is easily influenced by other documents or the LLM's internal knowledge, which is unreliable and lacks explainability. To address these limitations, we propose a **M**ask-**B**ased Membership Inference **A**ttacks (MBA) framework. Our framework first employs a masking algorithm that effectively masks a certain number of words in the target document. The masked text is then used to prompt the RAG system, and the RAG system is required to predict the mask values. If the target document appears in the knowledge database, the masked text will retrieve the complete target document as context, allowing for accurate mask prediction. Finally, we adopt a simple yet effective threshold-based method to infer the membership of target document by analyzing the accuracy of mask prediction. Our mask-based approach is more document-specific, making the RAG system's generation less susceptible to distractions from other documents or the LLM's internal knowledge. Extensive experiments demonstrate the effectiveness of our approach compared to existing baseline models.

## CCS CONCEPTS

• **Computing methodologies** → **Information extraction**; • **Security and privacy** → **Human and societal aspects of security and privacy**.

## KEYWORDS

Retrieval-Augmented Generation; Membership Inference Attacks

**ACM Reference Format:**
Anonymous Author(s). 2025. Mask-based Membership Inference Attacks for Retrieval-Augmented Generation. In *Proceedings of Make sure to enter the correct conference title from your rights confirmation emai (Conference acronym 'XX).* ACM, New York, NY, USA, 11 pages. https://doi.org/XXXXXXX.XXXXXXX

## 1 INTRODUCTION

Large language models (LLMs) such as ChatGPT [3] and LLama [34], have revolutionized natural language processing. Despite these advancements, challenges remain, particularly in handling domain-specific or highly specialized queries [16]. LLMs often resort to "hallucinations," fabricating information outside their training data [43].

Retrieval-Augmented Generation (RAG) addresses this by integrating external data retrieval into generation, improving response accuracy and relevance [11, 19]. And RAG has been widely adopted by many commercial question-and-answer (Q&A) systems to incorporate up-to-date and domain-specific knowledge. For instance, Gemini [33] leverages the search results from Google Search to enhance its generation, while Copilot[1] integrates the documents or pages returned by Bing search into its context.

A recent trend involves storing up-to-date or copyrighted data in RAG knowledge databases instead of using it for LLM training. The SILO framework [25] exemplifies this approach, training LLMs on low-risk data (e.g., public domain or permissively licensed) and storing high-risk data (e.g., medical text with personally identifiable information) in the knowledge base. However, the legal implications of using data for generation models or systems are under scrutiny, with lawsuits filed globally due to potential copyright infringement [6, 24, 29, 30, 35]. This concern has spurred the development of Membership Inference Attacks (MIAs) to detect if specific data records were stored in RAG's knowledge database and could potentially appear in the generated texts, which raises concerns about *fair use doctrine* [12] or *General Data Protection Regulation (GDPR)* compliance [42].

Even though a growing body of research has focused on enhancing the trustworthiness of RAG systems [26, 36, 41, 44, 45], to the best of our knowledge, there are only two existing works targeting at the MIAs in RAG system. RAG-MIAs [1] judges whether a target document is in the knowledge database by directly asking the RAG system (i.e., utilizing the RAG's response (yes or no) as the judgement). This approach relies solely on the RAG system's judgment, which is unreliable and lacks explainability. $S^2$MIAs [20] prompts the RAG system with the first half (typically the question part) of the target document, and if the RAG's response is semantically similar to the remaining half (typically the answer part) of the target document, the target document is judged as a member. Several studies have focused on Membership Inference Attacks (MIAs) for LLMs [4, 23, 39]. Among these, Min-k% Prob Attack [31]

---

[1]https://github.com/features/copilot/

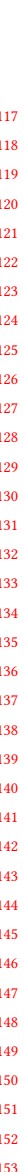
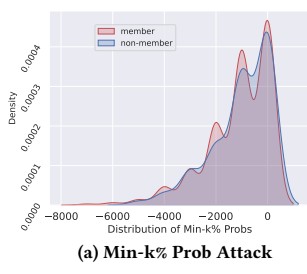
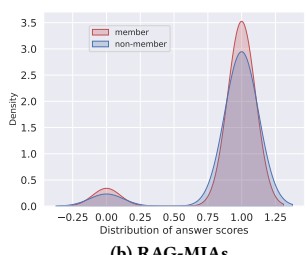
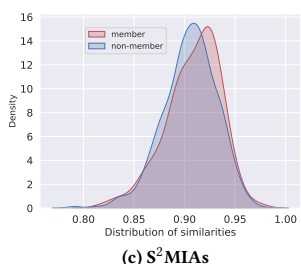
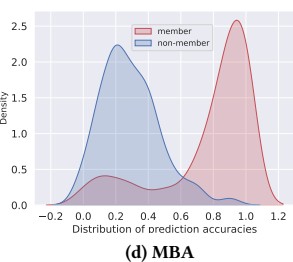

(a) Min-k% Prob Attack  (b) RAG-MIAs  (c) S²MIAs  (d) MBA

**Figure 1: Distributions of Indicators for Member and Non-Member Samples in Different Methods on HealthCareMagic-100k dataset, which are visualised by kernel density estimate (KDE) method.**

is a state-of-the-art method that infers membership using the sum of the minimum k% probabilities of output tokens. However, as illustrated in Figure 1 (a)-(c), the indicators used to determine membership in existing methods (e.g., the similarity between the second half of the target document and the generated response in S²MIA) are nearly indistinguishable for member and non-member samples. This hinders the effectiveness of MIAs in RAG systems.

To effectively and reliably detect whether a target document resides in a RAG system's knowledge database, we propose a **M**ask-**B**ased Membership Inference **A**ttacks (MBA) framework. The intuition is that if specific words (i.e., carefully selected words) in the document are masked, the RAG system is highly likely to predict the mask values accurately only if it retrieves the entire document as context. This prediction accuracy serves as our membership indicator. To conduct the inference, we first design a mask generation algorithm, masking $M$ words or phrases in the original target document, where $M$ is a hyperparameter. This involves extracting professional terms or proper nouns and selecting the most challenging words to predict using a pre-trained proxy language model. After obtaining the masked texts, we present the masked document to the RAG system and the RAG system is required to predict the mask values. A simple yet effective threshold-based judgement metric is adopted to determine the membership, i.e., if over $\gamma \cdot M$ masked words are correctly predicted, where $\gamma$ is a hyperparameter, we judge the target document as a member of the knowledge database. As shown in Figure 1 (d), compared to existing methods, our mask-based method exhibits a significant gap in mask prediction accuracy between member (avg. 0.9) and non-member (avg. 0.2) samples. This enables our approach to effectively and reliably determine the membership of the target document. We open source our code online[2].

To summarize, the main contributions of this paper are:

- We propose a Mask-based Membership Inference Attacks (MBA) framework targeting at the scenario of RAG system. Our framework is applicable to any RAG system, regardless of its underlying LLM parameters or retrieval method.
- We design a mask generation algorithm that strategically masks terms that would be difficult for the RAG system to predict if the full document were not retrieved as context.

---

[2]Code available after acceptance

- We evaluated our MBA framework on three public QA datasets. Extensive experiments demonstrated the effectiveness of our framework, achieving an improvement of approximately 50% in ROC AUC value compared to existing methods.

## 2 RELATED WORK
### 2.1 Retrieval-Augmented Generation

Retrieval-Augmented Generation (RAG) enhances response accuracy and relevance by incorporating external data retrieval into the generation process [11]. A common RAG paradigm involves using the user query to retrieve a set of documents, which are then concatenated with the original query and used as context [19].

Recent research has focused on various retrieval methods, including token-based retrieval [18], data chunk retrieval [28], and graph-based retrieval [9, 17]. Additionally, studies have explored adaptive retrieval [15] and multiple retrieval [14]. More advanced techniques such as query rewriting [10, 21] and alignment between retriever and LLM [5, 40] are beyond the scope of this paper.

### 2.2 Membership Inference Attacks

Membership Inference Attacks (MIAs) [13, 32] are privacy threats that aim to determine if a specific data record was used to train a machine learning model. MIAs for language models [4, 23, 31, 39] have been the subject of extensive research. Some representative attacking methods are: 1) **Loss Attack:** A classic MIA approach that classifies membership based on the model's loss for a target sample [39]; 2) **Zlib Entropy Attack:** It refines the Loss Attack by calibrating the loss using zlib compression size [4]; 3) **Neighborhood Attack:** This method targets MIAs in mask-based models by comparing model losses of similar samples generated by replacing words with synonyms [23]; 4) **The Min-k% Prob Attack:** this approach calculates membership by focusing on the k% of tokens with the lowest likelihoods in a sample and computing the average probabilities. However, these works may not be directly applicable to RAG systems. Additionally, many of them rely on the loss or token output probabilities, which require access to LLM parameters or intermediate outputs that may not be available in black-box RAG systems.

Recently, there are two works targeting at the MIAs in RAG scenarios. RAG-MIAs [1] judges whether a target document is in the

RAG system's knowledge database by prompting the RAG system with "*Does this: {Target Document} appear in the context? Answer with Yes or No.*", then utilizing the RAG's response (yes or no) as the judgement result directly. This approach relies solely on the RAG system's judgment, which can be unreliable and lacks explainability. $S^2$MIA [20] prompts the RAG system with the first half of the target document, and compares the semantic similarity of the RAG's response and the remaining half of the target document. To enhance robustness, that work also incorporates perplexity of the generated response. A model is trained to determine the threshold values for similarity and perplexity. Membership is judged based on these thresholds: a target document is considered a member if its similarity exceeds the threshold and its perplexity falls below the threshold. However, the RAG system may not always strictly adhere to the original texts, and responses generated using internal knowledge may have similar similarity scores to those generated with retrieved documents. This can lead to unreliable and unconvincing prediction results.

## 3 PRELIMINARIES

In this section, we establish the notation, provide a brief overview of Retrieval-Augmented Generation (RAG), and outline the specific task addressed in this paper.

### 3.1 RAG Overview

RAG systems typically consist of three primary components: a knowledge database, a retriever, and a large language model (LLM). The knowledge database, denoted as $\mathcal{D} = \{P_1, \cdots, P_N\}$, comprises a collection of documents sourced from authoritative and up-to-date sources. The retriever is a model capable of encoding both queries and documents into a common vector space to facilitate retrieval. The LLM, such as ChatGPT or Gemini, is a trained language model capable of generating text.

The RAG process unfolds as follows: Given a user query $\mathbf{q}$, the system retrieves $k$ relevant documents from $\mathcal{D}$ using the retriever:

$$\mathcal{P}_k = \text{RETRIEVE}(\mathbf{q}, \mathcal{D}, k) \tag{1}$$

Typically, retrieval is based on similarity metrics like inner product or cosine similarity. The retrieved documents, concatenated as $\mathcal{P}_k = [p_1 \oplus \cdots \oplus p_k]$, are then combined with a system prompt $\mathbf{s}$ and the original query to generate a response using the LLM:

$$\mathbf{r} = \mathbb{LLM}(\mathbf{s} \oplus \mathbf{q} \oplus \mathcal{P}_k) \tag{2}$$

Here, $[\cdot \oplus \cdot]$ represents the concatenation operation.

### 3.2 Task Formulation

We introduce the task of Membership Inference Attacks (MIAs) in RAG system.

**Attacker's Target:** Given a target document $d$, the objective is to determine whether $d$ is present in the RAG system's knowledge database $\mathcal{D}$.

**Attacker's Constraints:** We target at the black-box setting in RAG system. The attacker cannot access the RAG system's knowledge base ($\mathcal{D}$) or the LLM's parameters. However, they can interact with the system freely and repeatedly. The RAG system's response is solely textual, providing answers to the user's questions without explicitly displaying the contents of the retrieved documents. This scenario is realistic, as users typically have unrestricted access to chatbots.

**Attacker's Task:** The attacker's task is to design a **B**inary **M**embership **I**nference **C**lassifier (BMIC) that takes the target document ($d$), and the response of the RAG system ($\mathbf{r}$) as input. Formally, the probability of $d$ being in $\mathcal{D}$ is calculated as:

$$\begin{aligned} P(d \in \mathcal{D}) &= \mathbb{BMIC}(d, \mathbf{r}), \\ \mathbf{r} &= \mathbb{LLM}(\mathbf{s} \oplus Q_d \oplus \mathcal{P}_k) \end{aligned} \tag{3}$$

where $\mathbf{r}$ is generated by the LLM using a system prompt $\mathbf{s}$, a well designed question generated from $d$ (denoted as $Q_d$), and retrieved documents $\mathcal{P}_k$. Designing a method to generate $Q$ that can effectively differentiate between responses generated with and without the target document in the context is a key challenge in MIAs for RAG systems.

**MIA Workflow:** The MIA process involves generating questions based on the target document $d$. If the RAG system's response (answer) $\mathbf{r}$ accurately matches the original content of $d$, it can be inferred that $d$ is present in the knowledge database $\mathcal{D}$. Conversely, if there is a significant mismatch, it suggests that $d$ is not in $\mathcal{D}$.

**Designing Principals:** There are three main principals on designing the classifier and the adaption function:

(1) **Effective Retrieval:** $Q_d$ should successfully retrieve $d$ if $d \in \mathcal{D}$. Recall that in RAG, relevant documents are retrieved based on the user query and used as context for generation. In this context, $Q_d$ serves as the user query. If $d$ cannot be successfully retrieved, it implies that $d$ is not in $\mathcal{D}$, leading to a negative judgment.

(2) **Indirect Information:** $Q_d$ shall not directly reveal the information to be verified in the BMIC. While using $d$ directly as $Q_d$ might seem straightforward, it introduces bias: the RAG system will always include $d$ in the context, regardless of its presence in the knowledge base, making the inference unreliable.

(3) **Targeted Questions:** Questions should be challenging for the RAG system to answer if $d$ is not in the knowledge base, and vice versa. Overly simple questions can be answered using internal knowledge or other retrieved documents, hindering judgment. Conversely, irrelevant questions may not elicit expected responses, even if $d$ is successfully retrieved.

## 4 METHODS

### 4.1 Overview

This section presents our proposed **M**ask-**B**ased Membership Inference **A**ttacks (MBA) framework, illustrated in Figure 2. We begin by explaining our motivation (Section 4.2). Subsequently, we introduce the two key components of our framework: **Mask Generation** (Section 4.3), which generates $M$ masks within the original target document as our document-specific question ($Q_d$), and a **Binary Membership Inference Classifier** (BMIC, Section 4.4), which infers membership based on the masked texts. Our framework is non-parametric and can be applied to any black-box RAG system, regardless of LLM parameters or retrieval methods.

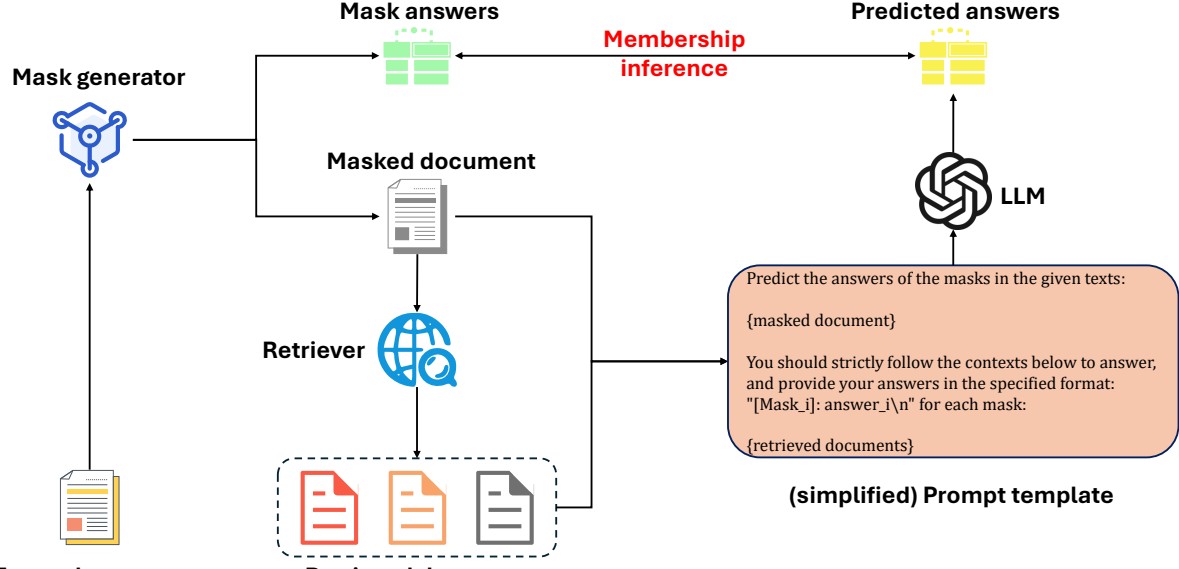

**Figure 2: Overview of the proposed MBA framework.**

## 4.2 Motivation

We observe that when LLMs are tasked with cloze tests (predicting masked terms or phrases in a given text), they can accurately fill in the blanks if the complete original text is provided as a reference. This phenomenon inspired us to conduct MIAs in RAG systems using a cloze test approach.

Specifically, the masked target document is used as a query to retrieve relevant documents from the knowledge database. Due to the high similarity between the masked document and the original one, the target document is highly likely to be retrieved if it appears in the knowledge database. In this case, the masked words can be accurately predicted. Conversely, if the target document is not in the database, there is no direct information to guide mask prediction, leading to inaccurate predictions. Therefore, the accuracy of mask prediction can serve as an indicator of the target document's membership.

## 4.3 Mask Generation

The first step involves generating masked text from the original target document, which acts as the document-specific question $(Q_d)$. We aim to select terms that are challenging to predict based solely on the LLM's internal knowledge or the context. While cloze question generation research [22, 38] exists, these works primarily focus on educational applications and may not be suitable for our purposes.

While it's tempting to use LLMs to generate masks, their inherent uncertainty presents two main challenges. First, LLMs may not always follow instructions to generate the desired number of masks. Second, the generated mask answers may not accurately align with the original words in the target document, potentially altering the document's content or even resulting in completely blank masked texts in some cases.

A straightforward approach would be to use a proxy language model to select terms based on their prediction difficulty (i.e., the probabilities to correctly predict them). However, we observed three challenges:

(1) **Fragmented words:** Datasets often contain specialized terms or proper nouns that may not be recognized by language model tokenizers. For example, GPT-2 might split "*canestan*" (a medicine) into "can," "est," and "an". Masking such terms based solely on prediction probability, which might generate "can*[Mask]*an", could hinder accurate prediction, even with the entire text retrieved.

(2) **Misspelled words:** Datasets collected from human-generated content may contain misspelled words (e.g., the word "nearly" is written as "nearlt"). If such words are masked, LLMs tend to accurately predict the correct spelling (e.g., "nearly"), despite prompted to follow the original text, affecting prediction accuracy.

(3) **Adjacent masks:** Masking two adjacent words can be problematic for LLMs. For instance, masking "walking" and "unsteadily" in the sentence "I went to the bathroom *[Mask_1]* (walking) *[Mask_2]* (unsteadily), as I tried to focus..." might lead the LLM to incorrectly predict "[Mask_1]: walking unsteadily; [Mask_2]: as I tried to focus". Specifically, the LLM might incorrectly identify the locations of the masked terms, despite its ability to effectively extract nearby terms or phrases.

To address these challenges, we incorporate an fragmented tokens extraction algorithm (Section 4.3.1), misspelled words correction (Section 4.3.2) and rule-based filtering methods into our mask proxy language model based generation process (Section 4.3.3).

*4.3.1 Fragmented words extraction.* We first extract words fragmented by the proxy language model's tokenizer, such as "canestan." The workflow involves identifying consecutive words (without spaces or punctuation in the middle) that are split by the tokenizer. This process is detailed in Algorithm 2 in Appendix A.1, where all the fragmented words are extracted and stored in the list fragmented_words.

*4.3.2 Misspelled words correction.* After extracting fragmented words, we further check whether they are misspelled words. If yes, their corrected words are also obtained and recorded, as detailed in Algorithm 1.

We iterate through each extracted fragmented word (lines 2-10). For the current word, we obtain its index in the target document (line 3). We use a pre-trained spelling correction model to check if the word is misspelled. We pass the current word and its preceding two words to the model and record the corrected word (lines 4-6). We empirically found that using three words provides the best results, as fewer words can lead to semantic inconsistencies. The corrected words are recorded along with the original misspelled words in the list fragmented_words (lines 7-9).

---

**Algorithm 1** WordsCorrection

**Input:** $d$
**Output:** fragmented_words
1: fragmented_words $\leftarrow$ FragmentedWordExtraction($d$)
2: **for** $i \in \{1, 2, \cdots, |\text{fragmented\_words}|\}$ **do**
3:     $index \leftarrow$ GETWORDINDEX$\left(\text{fragmented\_words}_{(i)}, d\right)$
4:     sub_sentence $\leftarrow \left[d_{(index-2)} \oplus d_{(index-1)} \oplus d_{index}\right]$
5:     corrected_words $\leftarrow \mathbb{SCLM}$ (sub_sentence)
6:     corrected_word $\leftarrow$ corrected_words$_{(3)}$
7:     **if** fragmented_words$_{(i)} \neq$ word **then**
8:         fragmented_words$_{(i)} \leftarrow \{\text{fragmented\_words}_{(i)},$ corrected_word$\}$
9:     **end if**
10: **end for**
11: **return** fragmented_words

---

*4.3.3 Proxy language model based masking.* To identify challenging words for masking, we employ a proxy language model. This model assigns a **rank score** to each word, reflecting its difficulty to predict.

Language models generate text by calculating the probability of each possible token given the preceding context and selecting the most likely one. The **rank score** indicates a word's position among these predicted candidates. For instance, in the sentence "... I would advise you to visit a [MASK] ...", if the correct token is "dentist" but the model predicts "doctor" (0.6), "medical" (0.25), and "dentist" (0.15), then "dentist" would have a rank score of 3. Words with *larger* rank scores are considered more challenging to predict. Therefore, we replace words with the largest rank scores (i.e., the words whose probabilities ranked at the back) with the "[MASK]" token, as detailed in Algorithm 4 of Appendix A.3.

The input $M$ represents the desired number of masks. We divide the target document into $M$ equal-length subtexts and distribute the $M$ masks evenly across these subtexts. For each subtext, we first add

the subtexts before the current subtext (i.e., $d_1$ to $d_{(i-1)}$) as a prefix, and iterate through its words, adding them one by one to the prefix. We then determine whether the next word should be masked based on several criteria: if it is a stop word, punctuation, or adjacent to an already masked word, it is not masked, which is implemented by assigning its probability rank as -1. If a word is eligible for masking, we use the proxy language model to calculate its probability of occurrence in the given context and record its **rank score**. For extracted fragmented words, we first check for misspelled errors. If found, we use the corrected word (obtained in Algorithm 1) for probability and **rank score** calculations. Otherwise, we calculate probabilities and **rank scores** for each token within the word, using the largest one to represent the word's overall **rank score**. For words that are not extracted fragmented words, their probabilities and **rank scores** are calculated by proxy language model directly.

The word with the largest **rank score** within each subtext is then masked, and its corresponding answer is recorded (lines 26-33). If the masked word is an misspelled word, both the original word and the corrected word will be added to the answer set (line 28).

*4.3.4 Mask integration.* Finally, the masked words are integrated and numbered. The "[Mask]" labels in the masked text are numbered from "[Mask_1]" to "[Mask_M]". A ground truth mask answer dictionary is maintained in the format "[Mask_i]: answer_i," where "answer_i" is the $i$-th masked word.

## 4.4 Binary Membership Inference Classifier

The RAG system is prompted with the template shown in Figure 6 in Appendix C.1, where the masked document is obtained using the method introduced in Section 4.3, and the {retrieved documents} represent those retrieved from the RAG's knowledge database.

The response will be in the format "[Mask_i]: answer_i," where "answer_i" represents the predicted answer for "[Mask_i]". We then compare the predicted answers with the ground truth answers and count the number of correct predictions. If this count exceeds $\gamma \cdot M$, where $\gamma \in (0, 1]$ is a hyperparameter, we judge the target document as a member of the RAG's knowledge database; otherwise, we conclude it is not a member.

## 5 EXPERIMENTS

### 5.1 Datasets

We evaluate our method on three publicly available question-answering (QA) datasets:

- **HealthCareMagic-100k**[3]: This dataset contains 112,165 real conversations between patients and doctors on Health-CareMagic.com.
- **MS-MARCO** [2]: This dataset features 100,000 real Bing questions with retrieved passages and human-generated answers. We use the "validation" set (10,047 QA pairs) for knowledge base construction. The knowledge base includes all unique documents retrieved by at least one question.
- **NQ-simplified**[4]: This is a modified version of the Natural Questions (NQ) dataset. Each question is paired with a

---

[3]https://huggingface.co/datasets/RafaelMPereira/HealthCareMagic-100k-Chat-Format-en
[4]https://huggingface.co/datasets/LLukas22/nq-simplified

shortened Wikipedia article containing the answer. We use the "test" set (16,039 QA pairs) to build a knowledge base by storing the shortened Wikipedia articles.

Following previous research [1, 20], we randomly selected 80% of the documents as member samples (stored in the RAG's knowledge base) and the remaining 20% as non-member samples. We randomly selected 1,000 instances for training (500 member and 500 non-member) and another 1,000 for testing (500 member and 500 non-member) to determine any necessary thresholds.

## 5.2 Baselines

We evaluated our method against the following baseline approaches:

- **Min-k% Prob Attack** [31]: A state-of-the-art membership inference attack (MIA) for LLMs. It calculates a score based on the sum of the least likely tokens to determine membership.
- **RAG-MIA** [1]: This method directly queries the RAG system about the target document's inclusion in the retrieved context.
- **$S^2$MIA** [20]: This approach divides the target document into two halves, prompts the RAG system with the first half, and compares the semantic similarity between the second half and the RAG's response. We compare 2 settings of $S^2$MIA:
  - $S^2$MIA$_s$: Relies solely on semantic similarity for MIA.
  - $S^2$MIA$_{s\&p}$: Incorporates both semantic similarity and perplexity for membership inference.

Of these methods, Min-k% Prob Attack and $S^2$MIA$_{s\&p}$ require token prediction probabilities, which may not be accessible in certain black-box settings.

## 5.3 Evaluation Metric

We evaluate performance using a comprehensive set of metrics. Notably, we introduce **Retrieval Recall** as a unique metric for MIAs in RAG systems, distinguishing our work from previous studies [1, 20]. Retrieval recall measures whether the target document is successfully retrieved from the knowledge base when it exists. If the target document is among the top $K$ retrieved documents, the recall is 1; otherwise, it is 0. We calculate the overall retrieval recall as the average across all membership documents, excluding non-member documents. In addition to retrieval recall, we also employ standard metrics commonly used in MIAs [8] and binary classification tasks, including **ROC AUC**, **Accuracy**, **Precision**, **Recall**, and **F1-score**. Specifically, member documents are labeled as 1, and non-member documents are labeled as 0. Each method outputs a logit value between 0 and 1 (e.g., the mask prediction accuracy), which is then used to calculate the metrics.

## 5.4 Settings and implementation

*5.4.1 General settings.* We leverage GPT-4o-mini[5] as our black-box LLM, which is accessed by OpenAI's API. For the RAG system, we utilize LangChain[6] framework and integrate BAAI/bge-small-en [37] as the retrieval model, which encodes both queries and

documents into 384-dimensional vectors. Retrieval is performed by calculating the inner product between these vectors, and an approximate nearest neighbor search is conducted using an HNSW index implemented in FAISS [7]. All experiments were conducted on a single NVIDIA RTX A5000 GPU.

*5.4.2 Method-specific settings.* We now detail the specific settings used in each method:

- **Min-k% Prob Attack:** k is a hyperparameter in this method. We varied k from 1 to 20, and selects the k with the best performance. This method also involves calculating the sum of minimum k% log probabilities as the indicator for membership inference. To obtain the log probabilities, we leverage the "logprobs" parameter within the OpenAI API[7].
- **$S^2$MIA:** Cosine similarity is used to measure the similarity between the second half of the original target document and the response generated by RAG. To calculate perplexity, the "logprobs" parameter is enabled to obtain the log probabilities of tokens, similar to the Min-k% Prob Attack method. XGBoost, as recommended in the original paper, is used as the binary classifier.
- **Our method:**
  - **Spelling Correction Model:** We leverage the pretrained "oliverguhr/spelling-correction-english-base" model (139M parameters) from Hugging Face[8] to address potential spelling errors.
  - **Proxy Language Model**: We employ the "openai-community/gpt2-xl" [27] model with 1.61B parameters as a proxy language model for difficulty prediction."
  - $M$: The number of masks is a hyperparameter in our method. We experimented with different values of $M$ in $\{5, 10, 15, 20\}$ for each dataset and selected the optimal $M$ that produced the highest ROC AUC value.
  - $\gamma$: The threshold for mask prediction accuracy, used to determine membership, is a hyperparameter in our method. We varied this threshold from 0.1 to 1 for each dataset and selected the optimal threshold ($\gamma$) that produced the highest F1-score.

The results obtained using the optimal $M$ are presented as our overall results.

## 5.5 Overall Performance

Table 1 presents the experimental results comparing our proposed MBA4RAG framework with baseline methods.

A key premise of MIAs in RAG is the successful retrieval of the target document if it exists in the knowledge database. Retrieval recall is therefore a crucial metric. Both RAG-MIA and MBA4RAG achieve high overall retrieval recall (over 0.9) due to their use of the full original target document or masked versions with high similarity. In contrast, $S^2$MIA and Min-k% Prob Attack retrieve documents based on fragments, leading to potential discrepancies, especially in chunked knowledge databases. These methods exhibit lower retrieval recall, particularly in the HealthCareMagic dataset, likely due to the similarity of many patient-doctor dialogues.

---

[5]https://openai.com/index/gpt-4o-mini-advancing-cost-efficient-intelligence/
[6]https://github.com/langchain-ai/langchain

[7]https://cookbook.openai.com/examples/using_logprobs
[8]https://huggingface.co/oliverguhr/spelling-correction-english-base

**Table 1: Performance comparison of different methods on MIAs for RAG systems.**

| Dataset | Model | Retrieval Recall | ROC AUC | Accuracy | Precision | Recall | F1-score |
|---|---|---|---|---|---|---|---|
| HealthCareMagic-100k | Min-k% Prob Attack | 0.65 | 0.38 | 0.60 | 0.75 | 0.75 | 0.75 |
| | RAG-MIA | **0.93** | 0.49 | 0.75 | 0.80 | 0.91 | 0.86 |
| | $S^2MIA_s$ | 0.62 | 0.46 | 0.77 | 0.79 | **0.96** | 0.87 |
| | $S^2MIA_{s\&p}$ | 0.62 | 0.57 | 0.78 | 0.85 | 0.92 | 0.89 |
| | MBA | 0.87 | **0.88** | **0.85** | **0.97** | 0.81 | **0.89** |
| MS-MARCO | Min-k% Prob Attack | 0.82 | 0.44 | 0.65 | 0.71 | 0.67 | 0.69 |
| | RAG-MIA | **0.98** | 0.52 | 0.75 | 0.81 | **0.90** | 0.85 |
| | $S^2MIA_s$ | 0.81 | 0.64 | 0.57 | 0.80 | 0.63 | 0.71 |
| | $S^2MIA_{s\&p}$ | 0.81 | 0.69 | 0.66 | 0.84 | 0.61 | 0.71 |
| | MBA | 0.97 | **0.86** | **0.81** | **0.91** | 0.85 | **0.88** |
| NQ-simplified | Min-k% Prob Attack | 0.81 | 0.65 | 0.58 | 0.79 | 0.68 | 0.73 |
| | RAG-MIA | 0.97 | 0.52 | 0.79 | 0.82 | **0.95** | 0.88 |
| | $S^2MIA_s$ | 0.81 | 0.67 | 0.64 | 0.89 | 0.64 | 0.74 |
| | $S^2MIA_{s\&p}$ | 0.81 | 0.68 | 0.66 | 0.87 | 0.68 | 0.76 |
| | MBA | **0.98** | **0.90** | **0.85** | **0.90** | 0.91 | **0.90** |

The rows in gray indicate models that require token log probabilities for calculations, which may not be accessible in certain scenarios. For each metric and dataset, the **best performance** is bolded, and the second-best is underlined.

**Table 2: The ablation study of our method.**

| Dataset | Model | Retrieval Recall | ROC AUC |
|---|---|---|---|
| HealthCareMagic-100k | Random | **0.88** | 0.68 |
| | LLM-based | 0.86 | 0.81 |
| | $MBA_{PLM}$ | 0.85 | 0.74 |
| | $MBA_{w/o SC}$ | 0.87 | 0.85 |
| | MBA | 0.87 | **0.88** |
| MS-MARCO | Random | **0.97** | 0.73 |
| | LLM-based | 0.95 | 0.80 |
| | $MBA_{PLM}$ | **0.97** | 0.76 |
| | $MBA_{w/o SC}$ | 0.96 | 0.84 |
| | MBA | **0.97** | **0.86** |
| NQ-simplified | Random | 0.96 | 0.75 |
| | LLM-based | 0.97 | 0.86 |
| | $MBA_{PLM}$ | 0.97 | 0.69 |
| | $MBA_{w/o SC}$ | **0.99** | 0.84 |
| | MBA | 0.98 | **0.90** |

For the specific performance, ROC AUC is a dominant metric for evaluating MIAs [1, 8, 20, 23]. Our method consistently outperforms baseline methods by nearly 50% across all datasets. Even though baseline methods may achieve notable performance on metrics like precision and recall, these results can be attributed to arbitrary strategies, such as judging all documents as members.

In conclusion, our mask-based MIA method effectively retrieves the target document when it exists in the knowledge database and focuses on the target document without being distracted by other retrieved documents. This leads to high performance and reliability.

## 5.6 Ablation Study

To assess the effectiveness of our mask generation method and its individual components, we compared it to several baseline approaches:

- **Random**: A simple baseline where masks are selected randomly.
- **LLM-based**: An alternative approach using an LLM to select words or phrases for masking. The prompt template is provided in Figure 7 in Appendix C.2.
- **$MBA_{PLM}$**: This variant only uses the proxy language model for word selection, omitting fragmented word extraction (Section 4.3.1) and misspelled word correction (Section 4.3.2).
- **$MBA_{w/o SC}$**: This variant excludes the misspelled word correction (Section 4.3.2) component from our full method.

The results are presented in Table 2. Due to the similarity between the masked text and the original text (with only a few words or phrases replaced), retrieval recall is generally high for all masking strategies. Even random masking achieves competitive performance (ROC AUC of around 0.7) due to the mask-based method's ability to resist distractions from other retrieved documents. However, random masking may generate masks for simple words (e.g., stop words), which can be easily predicted by the LLM, leading to false positives.

The LLM-based mask generation method is straightforward to implement and achieves acceptable performance in most cases (ROC AUC of about 0.8). However, due to the inherent uncertainty of LLMs, the original texts may be altered, and the number of generated masks may deviate from the desired amount.

Our proxy language-based mask generation method guarantees stable generation, ensuring exactly $M$ masks are generated and distributed evenly throughout the text. However, challenges such as fragmented words, adjacent masks, and misspelled words can hinder prediction accuracy. By incorporating fragmented word

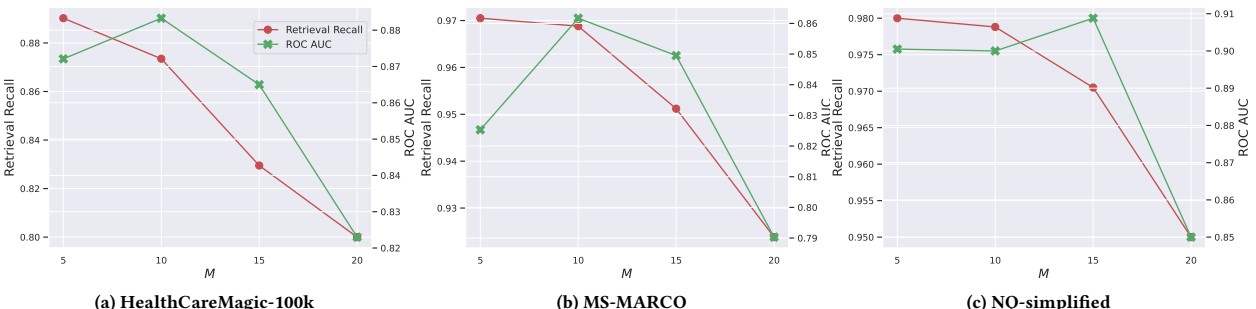

(a) HealthCareMagic-100k                    (b) MS-MARCO                    (c) NQ-simplified

**Figure 3: The performances comparison varying $M$**

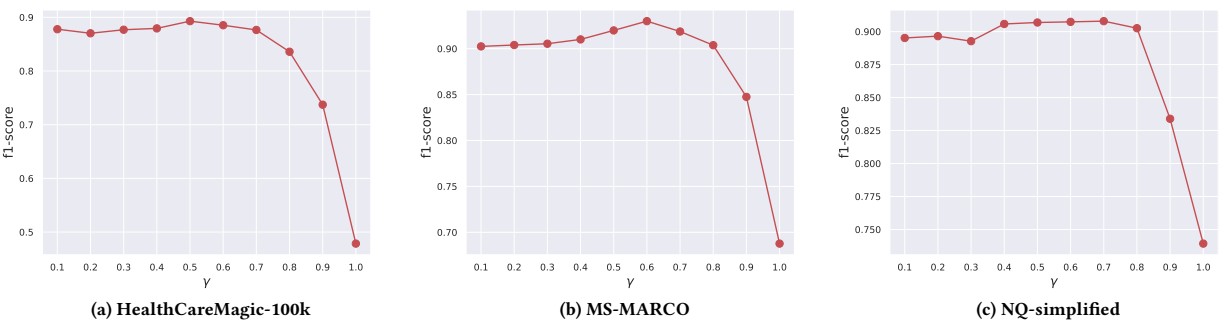

(a) HealthCareMagic-100k                    (b) MS-MARCO                    (c) NQ-simplified

**Figure 4: The performances comparison varying $\gamma$**

processing and misspelled word correction, our method achieves effective and reliable MIAs for RAG systems.

## 5.7 Parameter Study

*5.7.1 The impact of $M$.* To analyze the impact of $M$, the number of masks generated in the target document, we varied M in $\{5, 10, 15, 20\}$ and observed the retrieval recall and ROC AUC values (Figure 3).

As $M$ increases, retrieval recall slightly decreases. This is because more masked words reduce the similarity between the masked text and the original document. The ROC AUC value also fluctuates slightly. When $M$ is too small (e.g., below 5), the error tolerance decreases, meaning mispredictions have a larger impact on the final membership inference performance. When $M$ is too large (e.g., over 20), simple words may be masked, leading to accurate prediction without the target document being retrieved. Additionally, decreased retrieval recall can lower the prediction accuracy of member samples, impacting overall performance.

Therefore, setting $M$ between 5 and 15 (exclusive) is an optimal choice.

*5.7.2 The impact of $\gamma$.* To analyze the impact of the membership threshold $\gamma$, we varied $\gamma$ from 0.1 to 1.0. Since retrieval recall and ROC AUC scores are independent of $\gamma$ (as $\gamma$ does not affect mask generation), we focused on f1-scores.

Figure 4 illustrates the results. While the optimal $\gamma$ value varies across datasets (5, 6, and 7), performance is relatively consistent within the range of 0.5 to 0.7. This indicates that the performance is not highly sensitive to $\gamma$, and setting $\gamma$ around 0.5 is generally a good choice.

*5.7.3 The impact of $K$.* $K$ is a system parameter representing the number of retrieved documents in the RAG system, which may influence performance. However, this parameter is beyond our framework and inaccessible to users. We verified that our method is insensitive to K in Figure 5 of Appendix B.

## 6 CONCLUSION

In this paper, we address the problem of membership inference for RAG systems, and propose a **M**ask-**B**ased Membership Inference **A**ttacks (MBA) framework. Our approach involves a proxy language-based mask generation method and a simple yet effective threshold-based strategy for membership inference. Specifically, we mask words that have the largest rank scores as predicted by a proxy language model. The target RAG system would have most of the masks correctly predicted if the document is a member. Extensive experiments demonstrate the superiority of our method over existing baseline models.

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

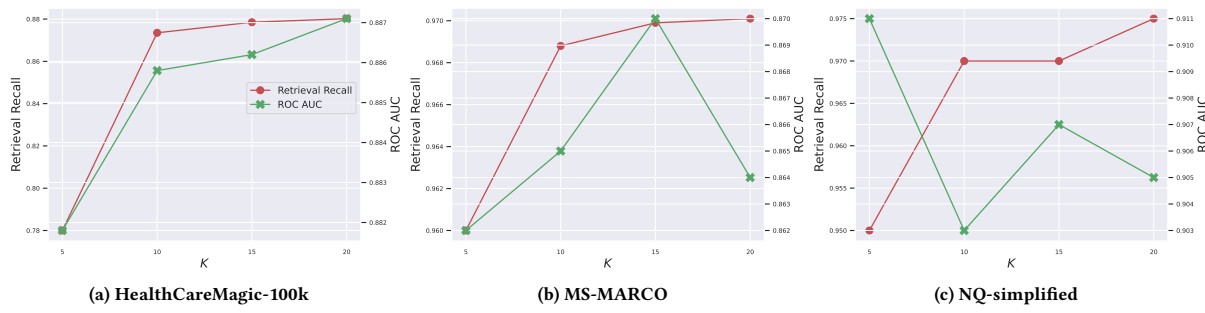

(a) HealthCareMagic-100k       (b) MS-MARCO       (c) NQ-simplified

**Figure 5: The performances comparison varying the number of $K$**

## A    MASK GENERATION ALGORITHMS

### A.1    The detailed algorithm of Fragmented words extraction

The workflow for extracting fragmented words is illustrated in Algorithm 2. It iterates through all words in d (lines 4-16). If the next word begins with a letter or certain hyphens, the words are combined into a single word.

---

**Algorithm 2** FragmentedWordExtraction

---

**Input:** $d$            ▷ the target document
**Output:** fragmented_words
1:   fragmented_words $\leftarrow \emptyset$
2:   $word \leftarrow ''$        ▷ set $word$ as an empty string
3:   $flag \leftarrow False$
4:   **for** $j \in \{1, 2, \cdots, |d| - 1\}$ **do**
5:     $word \leftarrow word \oplus d_j$
6:     **if** $d_{(j+1),0} \in \{[\text{a-z}],[\text{A-Z}],'\text{-}','\,/'\}$ **then**
7:       $flag \leftarrow True$
8:       continue
9:     **else**
10:      **if** flag **then**
11:        $flag \leftarrow False$
12:        Append(fragmented_words, $word$)
13:      **end if**
14:      $word \leftarrow ''$
15:     **end if**
16:   **end for**
17:   **return** fragmented_words

---

### A.2    The processing of fragmented words

As stated in Section 4.3.3, fragmented words are treated as a single unit. If a word is misspelled, the corrected word is used to calculate the probability and rank score (lines 1-4). Otherwise, all tokens within the fragmented word receive a rank score, and the lowest rank score represents the overall rank score of the entire fragmented word (lines 5-14).

---

**Algorithm 3** FragmentedWordsRank

---

**Input:** $d_i, j, t$
**Output:** $rank_{i,(j+1)}$
1:   **if** $\left[ d_{i,(j+1)} \oplus \cdots \oplus d_{i,(j+t)} \right]$ is unspelling **then**
2:     $d_{i,(j+1)} \leftarrow$ corrected_word
3:     $prob_{i,(j+1)} \leftarrow \mathbb{PLM}(d_{i,(j+1)} \mid sentence\_prefix)$
4:     $rank_{i,(j+1)} \leftarrow$ GETRANK($prob_{i,(j+1)}$)
5:   **else**
6:     $word \leftarrow ''$
7:     **for** $k \in \{1, 2, \cdots, t\}$ **do**
8:       $s' \leftarrow [sentence\_prefix \oplus word]$
9:       $prob_{i,(j+k)} \leftarrow \mathbb{PLM}(d_{i,(j+k)} \mid s')$
10:      $rank_{i,(j+k)} \leftarrow$ GETRANK($prob_{i,(j+k)}$)
11:      $word \leftarrow \left[ word \oplus d_{i,(j+k)} \right]$
12:     **end for**
13:     $rank_{i,(j+1)} \leftarrow \max_{k \in [1,t]} (rank_{i,(j+k)})$
14:   **end if**
15:   **return** $rank_{i,(j+1)}$

---

### A.3    Full algorithm of mask generation

The complete workflow of the mask generation is illustrated in Algorithm 4.

> You are given a text with several missing words or phrases, represented by placeholders in the format [Mask_i], where i is a unique number for each blank. Your task is to accurately fill in each placeholder with the most appropriate word or phrase based on the context of the sentence. Provide your answers in the specified format: "[Mask_i]: answer_i\n" for each mask, where "answer_i" shall be a word or phrase. You should strictly match the missing word or phrase based on the original context, without making any modifications, corrections, or substitutions.
>
> The text is: {target document}
> The context is: {retrieved document}

**Figure 6: The prompt template to predict the masked words.**

**Algorithm 4** MaskGeneration

**Input:** $d, M$
**Output:** $d_{Masked}$, answers
1: fragmented_words $\leftarrow$ WordsCorrection($d$)
2: $[d_1 \oplus \cdots \oplus d_M] \leftarrow$ SPLIT($d_{Masked}$) ▷ split into $M$ subtexts by length
3: $prefix \leftarrow \emptyset$
4: **for** $i \in \{1, 2, \cdots, M\}$ **do**
5:    $Prob\_Rank_i = \emptyset$
6:    $sentence\_prefix \leftarrow prefix$
7:    $s \leftarrow 0$
8:    **for** $j \in \{1, 2, \cdots, |d_i| - 1\}$ **do**
9:      $j \leftarrow j + s$
10:      $sentence\_prefix \leftarrow [sentence\_prefix \oplus d_{i,j}]$
11:      **if** $d_{i,(j+1)}$ is **stop word** or **punctuation** **then**
12:        $rank_{i,(j+1)} \leftarrow -1$
13:      **else if** $d_{i,(j)}$ or $d_{i,(j+2)}$ is "[Mask]" **then**
14:        $rank_{i,(j+1)} \leftarrow -1$    ▷ do no mask adjacent terms
15:      **else**
16:        **if** $\left[d_{i,(j+1)} \oplus \cdots \oplus d_{i,(j+t)}\right]$ in fragmented_words **then**
17:          $s \leftarrow s + t$
18:          $rank_{i,(j+1)} \leftarrow$ FragmentedWordsRank($d_i, j, t$)
19:        **else**
20:          $prob_{i,(j+1)} \leftarrow \mathbb{PLM}(d_{i,(j+1)} \mid sentence\_prefix)$
21:          $rank_{i,(j+1)} \leftarrow$ GETRANK($prob_{i,(j+1)}$)
22:        **end if**
23:      **end if**
24:      $Prob\_Rank_i \leftarrow$ Append($Prob\_Rank_i, rank_{i,(j+1)}$)
25:    **end for**
26:    $m \leftarrow$ argmax($Prob\_Rank_i$)    ▷ the token to be masked
27:    **if** $d_{i,m}$ is misspelled word **then**
28:      Append(answers, $\{d_{i,m},$ corrected_word$\}$)
29:    **else**
30:      Append(answers, $d_{i,m}$)
31:    **end if**
32:    Append(answers, $d_{i,m}$)
33:    $d_{i,m} \leftarrow$ "[Mask]"
34:    $prefix \leftarrow [prefix \oplus d_i]$
35: **end for**
36: $d_{Masked} = [d_1 \oplus \cdots \oplus d_M]$
37: **return** $d_{Masked}$, answers

---

You are given a text that needs {$M$} (strictly follow this number) words or phrases masked. Your task is to select words or phrases that would be challenging to guess if removed from the text and replace them with a placeholder in the format [Mask_i], where i is a unique number for each mask. Your answer shall be in the format of:
_Masked text_:
Provide the text with masks in place of the selected words or phrases.
_The answers for each mask_:
[Mask_1]: answer_1\n ... [Mask_{$M$}]: answer_{$M$}

**Figure 7: The prompt template to generate masks.**

## B THE IMPACT OF THE NUMBER OF RETRIEVED DOCUMENTS

We varied $K$, the number of retrieved documents, within the commonly adopted range of 5 to 20 in RAG systems. While increasing $K$ slightly improves retrieval recall, the ROC AUC value remains relatively constant. This indicates that our method is robust, and if the target document is retrieved, performance is guaranteed, regardless of the influence of other retrieved documents on mask prediction results.

## C PROMPT TEMPLATES

### C.1 Prompt template to predict mask answers

Figure 6 illustrates the prompt template used to predict mask answers based on masked texts. These predicted answers are then applied to conduct membership inference attacks as detailed in Section 4.4.

### C.2 Prompt template to generate masks

As demonstrated in the ablation study, an alternative to our proposed mask generation approach is to directly leverage the LLM. The prompt template for this direct approach is shown in Figure 7.

