# OpenReview forum: "Mask-based Membership Inference Attacks for Retrieval-Augmented Generation"
_ACM.org/TheWebConf/2025/Conference — WWW 2025 Poster_

### Official Review · Reviewer_1YFY · 2024-11-21

**Novelty:** 6
**Technical Quality:** 4

**Review:**

#### Summary
This study introduces the Mask-Based Membership Inference Attacks (MBA)
framework against RAG systems. The proposed mean infers membership by masking
a certain number of words in a target document based on the accuracy of predictions
for the masked (i.e., the document has been a member if the prediction was correct).
The evaluations demonstrate the effectiveness of MBA with three public QA datasets,
reaching around 50% ROC AUC compared to existing state-of-the-art MIAs against RAG.

#### Strengths
[+] Proposing a new attack vector in MIA with masking means
[+] Good performance with the datasets of the authors' choice

#### Weaknesses
[-] Insufficient description of a threat model
[-] Lack of discussions
[-] Lack of mitigations
[-] Lack of experiments on efficiency

Thanks for submitting the paper to WWW '25.

This paper is an interesting read. MIA on RAG is a crucial topic; hence I
believe the paper deals with it in a timely manner. I have several concerns as follows.

First, although Sec 3.2 describes the threat model of MBA (e.g., attacker's constraints),
it seems insufficient. The authors mentioned that MBA could be possible for
any RAG, but the effectiveness may differ depending on its implementations and policies.
For example, RAG can decide whether a query requires retrieval with query classification
to reduce the burden of retrieval. In such a case, the attacker cannot distinguish
between the results from LLM and those from RAG.

Second, similar to the first, a document may have been pre-processed for
a retrieval database. How does MBA affect under that configuration?
If one knows how MBA works, it is conceivable to evade the (copyright) detection
by properly pre-processing a document beforehand.
I guess this should be more explored in the future.

Third, including possible mitigation(s) for this attack would be great
(e.g., in a discussion section).

Fourth, the experiments on efficiency are missing.
How much computing resource does MBA require
compared to other state-of-the-art MIAs against RAG?

**Questions:**

- Have there been any experiments about the relationship between the effectiveness
  (e.g., F1) and the size of a document? This is a slightly different setting from adjusting M.
- What is the average size of a document (in terms of the number of tokens)?

**Reviewer Confidence:**

3: The reviewer is confident but not certain that the evaluation is correct

**Scope:**

3: The work is somewhat relevant to the Web and to the track, and is of narrow interest to a sub-community

---

### Official Review · Reviewer_jkdz · 2024-11-30

**Novelty:** 4
**Technical Quality:** 3

**Review:**

This paper presents a method for conducting membership inference attacks (MIAs) on Retrieval-Augmented Generation (RAG) systems, demonstrating a certain level of innovation and high writing quality. The paper concisely describes the process of masking words in the target document, prompting RAG to make predictions, and using a threshold-based inference method. While MIAs are a well-established concept, adapting them to RAGs with masking strategies introduces an inventive layer by addressing specific weaknesses in existing methods related to RAG’s management of up-to-date and domain-specific data.

Advantages:
1. The paper leverages a masking-based membership inference attack to explore the internal knowledge of RAG systems, showcasing a degree of innovation. It demonstrates superior attack performance compared to baseline methods.
2. The authors validate the effectiveness of the proposed method through extensive experiments.

Disadvantages:
1. The experiments only compare the attack method with baseline approaches, but the robustness of this MIA method against RAG defense strategies is not discussed.
2. The authors employ GPT-2-xl as a proxy language model to determine masked positions. However, given the substantial differences in parameter scale and training datasets between GPT-2-xl and GPT-4o-mini, GPT-4o-mini likely incorporates medical knowledge absent from GPT-2-xl. Thus, tokens that may be challenging for GPT-2-xl to predict could be relatively easy for GPT-4o-mini. Consequently, a potential limitation of this method is the inability to clarify whether the attack's success genuinely derives from the knowledge repository or merely from GPT-4o-mini's inherent capabilities.

**Questions:**

This paper presents a method for conducting membership inference attacks (MIAs) on Retrieval-Augmented Generation (RAG) systems, demonstrating a certain level of innovation and high writing quality. The paper concisely describes the process of masking words in the target document, prompting RAG to make predictions, and using a threshold-based inference method. While MIAs are a well-established concept, adapting them to RAGs with masking strategies introduces an inventive layer by addressing specific weaknesses in existing methods related to RAG’s management of up-to-date and domain-specific data.

Advantages:

1. The paper leverages a masking-based membership inference attack to explore the internal knowledge of RAG systems, showcasing a degree of innovation. It demonstrates superior attack performance compared to baseline methods.
2. The authors validate the effectiveness of the proposed method through extensive experiments.

Disadvantages:

1. The experiments only compare the attack method with baseline approaches, but the robustness of this MIA method against RAG defense strategies is not discussed.
2. The authors employ GPT-2-xl as a proxy language model to determine masked positions. However, given the substantial differences in parameter scale and training datasets between GPT-2-xl and GPT-4o-mini, GPT-4o-mini likely incorporates medical knowledge absent from GPT-2-xl. Thus, tokens that may be challenging for GPT-2-xl to predict could be relatively easy for GPT-4o-mini. Consequently, a potential limitation of this method is the inability to clarify whether the attack's success genuinely derives from the knowledge repository or merely from GPT-4o-mini's inherent capabilities.

**Reviewer Confidence:**

3: The reviewer is confident but not certain that the evaluation is correct

**Scope:**

2: The connection to the Web is incidental, e.g., use of Web data or API

---

### Official Review · Reviewer_MAdz · 2024-11-30

**Novelty:** 5
**Technical Quality:** 5

**Review:**

This paper introduces the Mask-Based Membership Inference Attacks (MBA) framework to improve Membership Inference Attacks (MIAs) in Retrieval-Augmented Generation (RAG) systems. MBA uses a masking algorithm to identify if a document is stored in the RAG system's knowledge database by analyzing the accuracy of mask predictions. Unlike previous methods, MBA is less influenced by external documents or internal LLM knowledge. Experimental results show that MBA outperforms existing models, offering a more reliable and document-specific approach to MIAs in RAG systems.

Strengths

**S1.** This paper introduces a mask-based Membership Inference Attack (MIA) for Retrieval-Augmented Generation (RAG) systems, a concept that is both understandable and logically sound. The explanation provided is insightful, particularly the observation that large language models (LLMs) can accurately restore phrases from reference texts during cloze tests, making the method interpretable.

**S2.** The paper appropriately references the two existing MIA attacks for RAG, and the baseline setup is well-justified.

**S3.** The presentation is smooth, the structure is clear, and the experiments are comprehensive. The innovative contributions are easy to follow and understand.

**S4.** The experimental results are significant, with notably high AUC scores.

Weaknesses

While the motivation of this work is clear, the proposed method appears straightforward and somewhat counterintuitive. Specifically, I have the following concerns:

**W1.** The challenges related to fragmented and misspelled words seem more like standard NLP tasks rather than substantial obstacles for MIA in RAG systems. Addressing them in a separate subsection may not be necessary.

**W2.** Moreover, "misspelled words" do not seem central to MIA for RAG systems. The algorithm should be robust to content variations, rather than focusing on word accuracy, especially for domain-specific terms.

**W3.** The MBA framework relies on a pre-trained language model to predict masked words. It is unclear whether this approach will produce significant differences across different proxy models or if the MBA's performance correlates with the match between the proxy model and the RAG system’s LLM.

**W4.** The performance of MBA depends on the selection of M and \gamma. For an adversary with limited prior knowledge and no ability to tune these parameters, achieving good performance may be difficult compared to existing methods.

**Questions:**

1. Could you compare token-level masking with word selection in MBA to demonstrate the importance of fragmented word masking?

2. Why not use simpler methods like TF-IDF for word selection? Could you provide a comparison?

3. Table 2 suggests that spell correction has minimal impact (AUC decreases by only 0.02 in MS-MARCO). Could you clarify this finding?

4. While MBA shows good AUC, would it be possible to include false positive rates for a more comprehensive evaluation?

**Reviewer Confidence:**

2: The reviewer is willing to defend the evaluation, but it is likely that the reviewer did not understand parts of the paper

**Scope:**

2: The connection to the Web is incidental, e.g., use of Web data or API

---

### Official Review · Reviewer_LbNY · 2024-12-01

**Novelty:** 3
**Technical Quality:** 3

**Review:**

### Summary
The paper proposes a Mask-Based Membership Inference Attack (MBA) for Retrieval-Augmented Generation (RAG) systems. It addresses the vulnerability of RAG systems to membership inference attacks, which aim to detect whether a document is included in the system's knowledge database. The proposed MBA framework masks certain words in a document and prompts the RAG system to predict these masked words. If the document is in the system, the complete document will be retrieved, allowing for accurate inference. The experiments on 3 datasets demonstrate its effectiveness.

### Strength
1. The idea is straightforward and easy to follow.
2. The method seems simple but effective based on the reported results.

### Weakness
1. The authors claim the source code is available, but I cannot access it.
2. Some of the writings in this paper are not clear. For example, in lines 255-256, the authors had better claim clearly why existing attacks are not applicable.
3. The experiments are not convincing. Many important experiments are mising. For instance, 1) how does MBA work in other backbone LLMs. 2) How about the impact of various RAG, retrieval models. 3) Will the differences in the mask words affect the attack performance?
4. The evaluation whether a sample is member or not highly depends on the hyperparameter $\gamma$.
5. Since the attack relies on the prediction accuracy of the masked word, it seems that a defender can rephase the text to easily bypass the attack.

Potential Limitation in Masking: The effectiveness of the mask-based approach might vary depending on how well the masking algorithm works across different document types or contexts.
Dependence on RAG System's Behavior: The method may still depend on how the RAG system retrieves or handles masked inputs, which might limit its robustness in real-world applications.
Generalizability: The framework might be specific to certain RAG systems, and its applicability to other types of AI models or databases is unclear.

**Questions:**

See the weaknesses.

**Reviewer Confidence:**

3: The reviewer is confident but not certain that the evaluation is correct

**Scope:**

3: The work is somewhat relevant to the Web and to the track, and is of narrow interest to a sub-community

---

### Official Review · Reviewer_gjj5 · 2024-12-02

**Novelty:** 5
**Technical Quality:** 5

**Review:**

This paper studies membership inference attacks (MIA) in RAG and proposes a mask-based membership inference attack (MBA) framework. The method introduced in the paper involves a proxy language-based mask generation method and a threshold-based membership inference strategy. Specifically, the author first masks the words with the maximum ranking score predicted by the proxy language model. If the retrieved document contains relevant information, the target RAG system will correctly predict most of the masks. Experiments have shown that the proposed method has an improvement of more than 20\% in ROC AUC value compared to the existing baseline model.

However, this method still has some shortcomings:
In the 4.3.3 Proxy Language Model Masking subsection, the author's intention seems to be to let the mask strategy select the most difficult words to predict, so as to test whether the target system can correctly predict these words. According to the following introduction, the author uses the gpt2 model as the proxy language model. There are two problems with this: On the one hand, there are many existing open source language models with different performance capabilities. The performance of different proxy language models (such as GPT-2 vs LLaMA 2) on the mask prediction task, the author did not analyze whether the new model has an impact on the experiment, such as the choice of mask words. On the other hand, the author assumes that words with high Rank Scores are the "hardest to predict words", but does not discuss the semantic background and actual meaning of these words in detail. Some rare words have low probabilities in most context predictions, and the author does not seem to point out the use of strategies such as top-k.
In addition, according to the experimental results, the effect of the Misspelled words correction proposed in this paper is not obvious.

Overall, the quality, clarity, originality and significance of this paper are acceptable.

**Questions:**

1. see review
2. The structural relationship of Fig2 is not clear, such as Mask answers and Masked document
3. Lack of case study

**Reviewer Confidence:**

4: The reviewer is certain that the evaluation is correct and very familiar with the relevant literature

**Scope:**

4: The work is relevant to the Web and to the track, and is of broad interest to the community